# Removal of Ibuprofen from Water by Different Types Membranes

**DOI:** 10.3390/polym13234082

**Published:** 2021-11-24

**Authors:** Mahdi Bourassi, Magda Kárászová, Mariia Pasichnyk, Raul Zazpe, Jana Herciková, Vlastimil Fíla, Jan M. Macak, Jana Gaálová

**Affiliations:** 1Institute of Chemical Process Fundamentals of the CAS, v.v.i., Rozvojova 135, 165 00 Prague, Czech Republic; bourassi@icpf.cas.cz (M.B.); karaszova@icpf.cas.cz (M.K.); pasichnyk@icpf.cas.cz (M.P.); 2Institute for Environmental Studies, Charles University, Benátská 2, 128 01 Prague 2, Czech Republic; 3Institut de Chimie des Milieux et Matériaux de Poitiers, 4 Rue Michel Brunet, TSA 51106, CEDEX 9, 86073 Poitiers, France; 4Center of Materials and Nanotechnologies, Faculty of Chemical Technology, University of Pardubice, Nam. Cs. Legii 565, 53002 Pardubice, Czech Republic; Raul.Zazpe@upce.cz (R.Z.); Jan.Macak@upce.cz (J.M.M.); 5Central European Institute of Technology, Brno University of Technology, Purkynova 123, 612 00 Brno, Czech Republic; 6Department of Organic Chemistry, University of Chemistry and Technology, Technická 5, 166 28 Prague 6, Czech Republic; Jana.Hercikova@vscht.cz; 7Department of Inorganic Technology, University of Chemistry and Technology, Technická 5, 166 28 Prague 6, Czech Republic; Vlastimil.Fila@vscht.cz

**Keywords:** ibuprofen, water treatment, dense polymer membrane, atomic layer deposition

## Abstract

Ibuprofen separation from water by adsorption and pertraction processes has been studied, comparing 16 different membranes. Tailor-made membranes based on Matrimid, Ultem, and diaminobenzene/diaminobenzoic acid with various contents of zeolite and graphene oxide, have been compared to the commercial polystyrene, polypropylene, and polydimethylsiloxane polymeric membranes. Experimental results revealed lower ibuprofen adsorption onto commercial membranes than onto tailor-made membranes (10–15% compared to 50–70%). However, the mechanical stability of commercial membranes allowed the pertraction process application, which displayed a superior quantity of ibuprofen eliminated. Additionally, the saturation of the best-performing commercial membrane, polydimethylsiloxane, was notably prevented by atomic layer deposition of (3-aminopropyl)triethoxysilane.

## 1. Introduction

Separation of residual pharmaceuticals and personal care products from wastewater has become a widely discussed contemporary research topic. Ibuprofen (IBU), in particular is one of the most consumed non-steroidal anti-inflammatory drugs. IBU shows some specific properties (see Table 1), such as chirality or hydrophobicity, linked to its acidic character and molecular structure. All these characteristics may have a huge influence on the quality of separation [1].

IBU with low logKow values may be considered relatively hydrophilic and tending to show high water solubility. That is why the concentration of residual IBU and its metabolites in waters increases dramatically [1]. IBU was detected both in groundwater [2] and drinking water [3]. It was noted that the concentration of some pharmaceuticals like diclofenac [4] and verapamil [5] could be effectively reduced during their passage through tertiary wastewater treatment plants. In contrast, IBU is sufficiently persistent to occur in estuarine systems [6].

For example, the report of water management in the Czech Republic shows that the concentrations of IBU and its major metabolites hydroxyibuprofen and carboxyibuprofen in surface water were 6.1 μg/L in 2015 [7] and 21 μg/L in 2018.

Membrane separations are well known in water treatment [8]. Up to now, processes such as ultrafiltration [9], nanofiltration [10], or reverse osmosis [11] are commonly used for the elimination of pharmaceuticals from water. The treatment can be enhanced by electrochemical oxidation of the reverse osmosis concentrate [12]. The removal of different pollutants depends on membrane thickness and composition, its compatibility with the pollutants, and experimental conditions [13]. In the case of IBU, its negatively charged nature, explains its solubility in water at defined pH conditions [14]. However, the hydrophobic character should not be ignored, the electrons from the π-π bond can interact with neutral surfaces [15,16]. Previous studies stated the membrane adsorption-removal mechanism effectiveness for relatively hydrophobic compounds [17,18].

The authors published in 2020 a review evaluating advantages and disadvantages of well-known membrane separation materials based on particle size, usually exposed to a large amount of water, versus dense hydrophobic membranes with targeted transport of contaminants through a selective barrier [19,20,21]. Electrodialysis (ED) with porous membrane has shown interesting membrane separation applications [22,23] in terms of valuable compound recovery from wastewater [17,24,25,26]. However, there are many challenges to overcome for a real full-scale application such as adsorbed pollutants on an ionic membrane, weak separation of macromolecules due to excursive membranes [24], low productivity and energy demanding, polarisation and fouling phenomena [25], and complex experimental setup (electrode, cationic and anionic membranes, and elution solution). Some other issues should be considered concerning limitations of neutral molecules and bipolar molecules, e.g., pH adaptation, water ionization, and target pollutant degradation at high current [26]. Osmosis processes (forward and reverse osmosis) are widely used for water desalination and water demineralization [27]. Using semi-permeable membranes such processes have known successful water treatment [28]. Nevertheless, these processes suffer from some issues to overcome such as quick fouling, requiring high pressure for reverse osmosis, and high concentrated draw solution for forward osmosis. Regardless of membrane separation process technique, membrane characteristics have a major effect [29], especially on membrane-pollutant interactions [19]. To investigate this interaction, pertraction [30] allows following separation process out of any interfering agent [31] (pressure, temperature, draw solution, current, etc.) as highlighted in our previous works [32].

The novelty of the proposed approach to water purification presented in this work consists of the use of hydrophobic membranes rather than hydrophilic ones, in a pertraction or pervaporation process. To date, hydrophobic membranes are better known in gas purification, but have been scarcely studied for the decontamination of aqueous media. Therefore, herein an extensive study of the performance of 16 different membranes (commercial and tailored membranes) towards IBU elimination is presented. Thus, the aim of this study was to select/prepare the most suitable membrane for the elimination of IBU from water from the materials studied in the laboratory of the research team. Following, based on the results obtained, to develop this new way of IBU elimination. The research focused on the best commercial membranes, as well as the tailor-made ones, previously studied in the pervaporation of diethyl phthalate [33]. The tailor-made materials containing different fillers—zeolites [34], graphene oxide [35], and single-walled carbon nanotubes (SWCNTs) [36] have shown high adsorption capacity to IBU.

The filler materials of the tailor-made membranes were chosen for their expected interactions between the permeating component and the membrane. Physicochemical properties like membrane charge characteristics [37] and hydrophobicity, water quality conditions like solution pH and ionic strength [38], membrane properties, and operating conditions strongly influence the removal of IBU [39]. At pH values below the IBU pKa value, it was shown that IBU was adsorbed and partitioned through the membrane [40]. It also must be noted that salts have a significant influence on the retention of IBU by membrane [41].

Regarding the commercial membranes, they were used as fabricated, or modified. The modification of the polydimethylsiloxane (PDMS) membrane surface with (3-aminopropyl)triethoxysilane molecules to increase the hydrophobicity nature of the membrane surface was also explored by modified atomic layer deposition (ALD). This is a well-established deposition method based on alternating gas–surface self-limited reactions that enable a conformal and uniform coating or decoration of different nanostructures, including one-dimensional nanopores and nanotubes [42,43]. In contrast to the usual ALD process, where a given precursor and corresponding co-reactant react in a self-terminating mode on the surface, we applied only half ALD cycles with no co-reactant. Accordingly, in order to essentially modify the surface of the PDMS membrane surface, a silane agent was repetitively dosed using a traditional ALD sequence, i.e., pulse and purge steps. This approach, previously presented in a recent work and referred to as a modified ALD approach [44], led to the desired surface modification of PDMS membranes with silane molecules.

The tests of the sorption capacity and the pertraction itself were completed by characterisations of the separation materials, which explained their dissimilarities during IBU elimination.

## 2. Materials and Methods

### 2.1. Materials

#### 2.1.1. Commercial Membranes

Polydimethylsiloxane (PDMS) membrane PERVAP 4060 was purchased from DeltaMem AG (Allschwil, Switzerland). PDMS is a thin-film composite membrane consisting of a polydimethylsiloxane thin layer supported on polyethylene terephthalate (PET). The membrane is commonly used for organophilic pervaporation. Polystyrene (PS) and polypropylene (PP) were clear dense polymeric films purchased from GoodFellow Cambridge Ltd. (Huntingdon, UK). The corresponding features of these commercial membranes are summarised in Table 2.

#### 2.1.2. Tailor-Made Membranes

Table 3 displays the different tailor-made membranes along with the corresponding materials and different fillers explored. Detailed description of such membranes and the corresponding fabrication process can be found in works previously published [45,46,47,48].

The preparations of Matrimid and 6FDA-DAM-DABA based membranes were published previously in works [48,49], respectively. The polyetherimide/GO membranes were prepared by the solution casting method. Firstly the solutions of 10 wt. % of polyetherimide (Ultem 1000, Sabic, Riyadh, Saudi Arabia) in *N*,*N*-Dimethylacetamide (SigmaAldrich, Prague, Czech Republic) with appropriate amounts of graphene oxide (0, 0.5, 2.5, and 5 wt. %) were prepared. Before casting onto level glass plates, the solutions were further sonicated for 1 h at laboratory temperature. For the casting, the stainless-steel knife with a gap of 1 mm was applied. The glass plates with casted mixtures were placed for 24 h into dust-proof Petri dishes in an oven and the solvent was evaporated at 50 °C. The peeled-off membranes were consequently vacuum dried for a few days at room temperature, to remove the residual solvent.

The slightly hydrophobic character of tailor-made membranes Ultem [50], 6FDA-DAM:DABA [51], and Matrimid [52] was earlier reported. The surface modification with GO [53] and TS-1 showed ability to increase the hydrophobicity [54].

#### 2.1.3. Materials Used for Atomic Layer Deposition

The surface of aforementioned PDMS membranes was modified with silane molecules by a modified ALD approach using the silane agent (3-aminopropyl)triethoxysilane > 98% (Sigma-Aldrich, Taufkirchen, Germany).

#### 2.1.4. Atomic Layer Deposition Silane-Modified PDMS Membrane

The PDMS membrane surface was modified with adsorbed (3-aminopropyl)triethoxysilane molecules using a modified ALD approach (thermal ALD, TFS 200, Beneq, Espoo, Finland). The modified ALD processes were carried out at a deposition temperature of 70 °C and a chamber pressure of 2 mbar. The (3-aminopropyl)triethoxysilane agent was heated to 68 °C to obtain a sufficient vapour pressure. Thus, one modified ALD cycle was defined by the following sequence: (3-aminopropyl)triethoxysilane agent (2 s) followed by N_2_ purge (45 s). The total number of cycles was 200. A blank reference membrane was prepared by applying the same thermal and atmospheric ALD process, except the exposure to the silane agent. All processes used N_2_ (99.999%) as a carrier gas at a flow rate of 400 cm^3^/min.

### 2.2. Methods

#### 2.2.1. Sample Preparation

The stock IBU solution for all experiments was prepared by adding 50 mg of IBU to 1 L of ultrapure water and stirring at 200 rpm for 12 h at room temperature overnight. The samples taken from pertraction and sorption experiments had volumes of 1.5 mL and were not measured directly. Instead, they were dried under a N_2_ atmosphere at room pressure and temperature. The samples for analysis (from Feed and Permeate) were dried and dissolved in 1/1 volume ratio acetonitrile/methanol in the way to be clearly detectable by HPLC detector.

#### 2.2.2. Sorption Measurements

The preferential sorption of IBU is depicted in Figure 1. The experiments were performed using dark glass bottles of 50 mL volume containing concentrated IBU solution (50 mg/L) and the tested membrane (cut into discs of diameter 2.1 cm, then by scissors into four similar pieces to ensure the maximal contact of the membrane with the solution). The bottles were placed on mechanical shaker and stirred at 130 rpm.

Sorption duration experiments varied depending on membrane material and on results obtained for previously collected samples. The experiments were given enough time for the membrane to adsorb the pollutant and to reach adsorption-desorption equilibrium. Samples of 1.5 mL volume were taken by micropipette after 30 min, 1 h, 3 h, and 1 day and then as needed.

#### 2.2.3. Pertraction Measurement

Pertraction experiments were performed in a closed, circular stainless-steel cell of 5.8 cm diameter and 6 cm length. The cell consisted of two compartments between which the membrane was held between two stainless-steel discs. The layout of the cell is shown in Figure 2 within the scheme of the pertraction set-up.

A constant temperature of 25 °C was assured by a recirculating cooler/chiller pumping water through the double wall of the cell. The membrane was cut to the desired size immediately before the experiment into round shape of 3 or 3.8 cm diameter, then fixed between two parts of the disc with screws and the leaking test has been provided. The cell was then closed from both sides and the chambers were filled simultaneously with deionised water (acceptor phase) and IBU solution (300 mg/L; donor phase), so that the pressure was kept equal on both sides of the membrane. Both chambers were equipped with a glass-coated magnetic stirrer and stirred constantly using external rotating magnets. The samples from both compartments were taken through septa by disposable sanitary syringes (1.5 mL) and analysed by HPLC. The commercial membranes, with properties described in Table 2, were used for pertraction. The pertraction process assesses the ability of a membrane to diffuse the selected pollutant from the feed to the receiving solution. Different parameters, such as temperature, feed and receiving solutions (water or organic solutions), stirring, intensity, membrane parameters (area, thickness, and materials), and pollutant selected are adjustable. In our experiment, both feed and receiving solutions were based on ultrapure water.

#### 2.2.4. Analytics

All samples taken during sorption and pertraction were analysed twice for reproducibility by HPLC on an 1100 Series HPLC from Agilent Technologies equipped with a binary pump, degasser, photo diode array (PDA) detector, solvent tray, and an auto sampler set to 10-µL injection. An analytical column CHIRALPAK^®^ QN-AX [*O*-9-(tert-butylcarbamoyl) quinine] (15 cm) was used. The detection wavelength was set to 230 nm where is a maximum absorption by IBU. We developed a method for HPLC measurement using a mobile phase composed of 50% acetonitrile and 50% methanol with 50 mM formic acid and 25 mM diethylamine as buffers. The temperature was kept at 25 °C and flow rate at 0.6 mL/min. Using the calibration curve, concentrations of IBU were determined.

#### 2.2.5. Characterisation Methods

The composition of the PDMS before and after the ALD process was monitored by XPS (ESCA2SR, Scienta-Omicron, Taunusstein, Germany) using a monochromatic Al Kα (1486.7 eV) X-ray source. The binding energy scale was referenced to adventitious carbon (284.8 eV). The quantitative analysis was performed using the elemental sensitivity factors provided by the manufacturer.

The pH of the point of zero charge was measured using the pH drift method. The pH of the polymer films in 0.01 M NaCl solution was adjusted to between 2 and 12 by adding 0.01 M NaOH or 0.01 M HCl. Polymer film (0.02 g) was added to 5 mL of the solution. The final pH was measured after 24 h.

The sessile-drop water contact angle measurements were carried out using an OCA 20 instrument (Dataphysics Products, Filderstadt, Germany). A drop volume of 2 µL was used and the average of six readings was taken. Surface elemental composition by X-ray photoelectron spectroscopy (XPS) of the unmodified and modified membranes was performed using an Escalab 250 spectrometer (Thermo Fisher Scientific, Waltham, MA, USA). To achieve as accurate result as possible, five simultaneous experiments were carried out with commercial membranes. The presented results are the corresponding average values.

## 3. Results

### 3.1. Sorption

#### 3.1.1. Sorption of IBU from Water by Tailor-Made Membranes

One set of measurements of IBU sorption from water was performed using the tailor-made membranes based on Matrimid, 6FDA-DAM:DABA 3:1, and Ultem polymers. The duration of the experiments was extended to observe IBU membrane-pollutant sorption kinetics and adsorption/desorption equilibria. Figure 3, Figure 4 and Figure 5 depict the time-dependence of the IBU concentration in the feed solution [55].

Figure 3 shows the 70-day-duration measurement of sorption of IBU by pure 6FDA-DAM:DABA 3:1, as well as 6FDA-DAM:DABA 3:1 with two different loading of graphene oxide (GO): 2.8 and 5 wt. %. The figure shows the maximal uptake of IBU by these separation materials up to 50% of initial IBU concentration. The addition of GO influences the rate of sorption and desorption of IBU to only a small extent. All three membranes show similar behaviour: the maximal uptake of IBU between 45–50% of initial IBU. However, there are oscillations of sorption/desorption of IBU visible for all three materials, increasing with the amount of GO in the membranes. The addition of 2.8 wt. % of GO increases the amplitudes of desorption and re-adsorption compared to pure 6FDA-DAM:DABA 3:1 (Figure 3). The effect is even more marked in the case of 5 wt. % GO loading, where the sorption capacity decreases. A higher loading may lead to the formation of aggregates of graphene oxide in the membrane, which results in higher disturbance in IBU sorption/desorption equilibrium and prevents sorption [56,57].

Figure 4 shows the results of similar experiments performed with Matrimid-based mixed matrix membranes whereas filler the titanium silicate (TS-1) zeolite with Si/Ti ratio of 25 has been used. The used low Si/Ti ratio compared to higher Si/Ti ratio enhances the compatibility between TS-1 particles and polymer, which results in more stable membranes [45]. The enhancement of compatibility is due to the presence of nanoparticles of TiO_2_ at the surface of TS-1 [58] promoting a better adhesion between particles and the carbonyl group from Matrimid [45].

The slowest and lowest sorption was observed for pure Matrimid. The addition of 10 wt. % TS-1 zeolite increased the maximum IBU uptake of Matrimid (Figure 4). This effect doubled by the addition of 20 wt. % TS-1 zeolite. However, 30 wt. % TS-1 zeolites in Matrimid turned out to be too high to increase the effect and 20 wt. % TS-1 zeolite in Matrimid turns out to be the optimal fraction to enhance membrane sorption capacity. The decreased sorption capacity of the membrane by increasing the TS-1 content over the optimal one, can be a consequence of TS-1 particles aggregation and an increase of the number of defects (polymer rigidification, cage-like structure) at particle-polymer interface [59], which decreases the accessibility of the TS-1 particle surface for IBU adsorption.

Figure 5 compares the rates of sorption/desorption of IBU by Matrimid membranes loaded with two different types of zeolite: TS-1 and ETS-10. Both membranes contained 30 wt. % of the zeolite. Zeolite ETS-10 Matrimid renders slightly better in the sorption of IBU compared to TS-1 Matrimid, but it is less stable in terms of sorption/desorption equilibrium. This is attributed to the presence of sodium and potassium cations in the ETS-10 structure [60].

Figure 6 shows the sorption of IBU by the Ultem based membrane, pure and containing different weight percentages of GO (0.5, 2.5 and 5 wt. %). Here, similar to 6FDA-DAM:DABA 3:1, the addition of GO slightly influences the sorption as well as desorption of IBU. Nonetheless, all membranes show similar behaviour, with the maximal uptake of IBU around 50% of initial IBU. Similar to DAM/DABA membranes, the Ultem materials show a certain optimum of GO filling, close to 2.5 wt. %. The Ultem membrane containing 2.5 wt. % GO shows the best sorption capacity of all tested tailor-made membranes, by adsorbing 70% of IBU in 12 days. The results are in good accordance with the literature [61].

#### 3.1.2. Sorption of IBU by Commercial Membranes

Compared to tailor-made membranes, the commercial membranes showed a better tendency to reach sorption equilibrium; however, they displayed a smaller capacity to eliminate IBU from water solution by sorption (10–15% compared to 50–70%). PP and PS are low-density homogeneous materials, and all commercial membranes floated on the surface of the solution during sorption. On the other hand, the PDMS membrane is a composite material consisting of a selective polydimethylsiloxane layer and porous support (PET). Figure 7 shows the sorption of IBU of such commercial membranes, revealing the best performance corresponded to the PMDS membrane. Both sides surfaces of the membrane were in contact with the solution.

The comparison of maximum sorption capacities in IBU elimination from water is shown in Figure 8 for 15 tested membranes. Plainly, the highest capacity is shown by Matrimid + 20 wt. % TS-1 zeolite, 0.58 mg/cm^2^. The second best membrane is Ultem + 2.5 wt. % GO (0.41 mg/cm^2^) and the third is 6FDA-DAM:DABA 3:1 + 2.8 wt. % GO (0.38 mg/cm^2^). An optimal amount of the filler is seen for each type of tailor-made polymer membrane. The commercial membranes are significantly weaker adsorbents, but they are mechanically more stable, user-friendly, and readily available. The tailor-made membranes are often brittle, fragile, and easily damaged during the experiment [62]. For this reason, a comparison of tailor-made membranes could not be achieved by the pertraction process; however, the present results form the basis for subsequent research in the field.

### 3.2. Results of Pertraction Measurements

For the above reasons, the authors focused on commercial membranes for the pertraction of IBU from water. Figure 9 shows the evolution of IBU concentration in the donor and acceptor phase during the pertraction process when applying PDMS, PS, and PP membranes.

Figure 9 reveals that the PDMS membrane was clearly able to permeate IBU from acceptor to donor phase. Given the novelty of this elimination of IBU from water, the high and predictive selectivity of this dense membrane (difficult to achieve with a porous membrane), it was a great success. Its performance is most probably due to the thin layer of PDMS material. IBU diffused to an insignificant extent through the PS and PP membranes, yet a small increase of concentration in acceptor phase obviously occurred. The IBU, however, stayed mostly adsorbed to these membranes; the diffusion of IBU through the PDMS membrane was much higher than that for PS and PP [63]. The accumulation of IBU in the membrane can be calculated from the pertraction measurement data, using mass balance between donor and acceptor phase. The calculated uptakes per cm^2^ of the membrane are presented in Figure 10. Evidently, during pertraction, the adsorption capacity of all membranes doubled or tripled.

#### 3.2.1. Characterisation of an ALD-Modified PDMS Membrane and Its Use in the Pertraction of IBU from Water

PDMS membranes happen to be the best commercial membranes for IBU pertraction. Mechanically very stable, accessible membranes have been modified by ALD as described previously. A quantitative analysis of PDMS and ALD-modified PDMS membranes is given in Table 4.

When discussing the XPS results (Table 4), we can summarise that a significant increase in Si content was observed after silane ALD, and a decrease in N content was caused by shading effect due to the silane layer on the surface. According to the results, we can suggest that the modified ALD approach led to the deposition (adsorption) of silane molecules on the membrane surface.

ALD modification of the PDMS membrane also results in a slight increase in water contact angle (WCA) originated by a decrease in membrane wettability. The results, shown in Figure 11, display the change in WCA with time. While the WCA of unmodified PDMS membrane decreases rapidly from its original 80° to 30°, the WCA of the ALD-modified counterpart is more stable, decreasing from 120° to 90° after 500 s.

Pertraction of IBU using the ALD-modified PDMS membrane took place in a very similar way during the first week (Figure 12) to the unmodified counterpart. The ALD-modified membrane was able to permeate more than 15% of IBU and absorbed up to 70 µg/cm^2^ of IBU. However, thereafter the PDMS membrane without modification stopped adsorbing IBU from the donor phase. The ALD membrane modification significantly improved the capacity of separation material to adsorb IBU compared to bare PDMS (53% instead of 23% of IBU in 16 days).

#### 3.2.2. Point of Zero Charge

In order to determine the charge of the adsorbent surface, the point of zero charge (pzc) was determined (Figure 13). The pH value at the pcz of the unmodified PDMS membrane is 4.16; at pH-values above pzc, the surface is negatively charged. The surface charge of PDMS in an aqueous medium is linked to the presence of Si-OH groups. The ALD-modified membrane has a higher pH value at the pzc: 5.78. This is due to the presence of NH_2_ groups as SiO_2_–NH_2_ [64]. Therefore, it carries a slight negative surface charge in basic conditions, which can explain the increasing sorption capacity towards IBU.

## 4. Discussion

The tested membranes, consisting of various polymers, demonstrated different degrees of sorption, permeation, and desorption of IBU. The main differences in IBU removal can be explained by the physical and chemical properties of the membranes. There exists a clear connection between the molecular structure of IBU and the chemical interaction with a membrane. The chemical structure of IBU is characterised by one centre with negative potential. The membrane surface carries various active groups; knowledge of charge distribution helps to clarify its adsorption capacity.

Commercial PDMS membrane PERVAP 4060 is the thinnest, resulting in a low enrichment factor and linked to the ratio of IBU concentration in the permeate to that in the feed. The existence of the α,ω-dihydroxypolydimethylsiloxane layer creates an electrostatic attraction towards the ionised forms of IBU, which explains the low adsorption capacity of the membrane. In the pertraction experiment, permeation is more favourable for the electrically neutral form of IBU and not strong ionic interaction between positively charged membrane and negatively charged IBU. The IBU molecule is not held on the surface of the membrane but passes through to the acceptor phase (Figure 1).

The Matrimid membrane modified by 20% TS-1 zeolite exhibited the greatest adsorption capacity. This membrane can accumulate the greatest amount of IBU compared to all other commercial or tailor-made membranes. TS-1 is a zeolite, in which a small number of silicon atoms in the framework are substituted for tetravalent Ti^4+^ atoms. Ti is positively charged in this compound, and it can readily interact with negatively charged IBU by dipole–dipole forces. Furthermore, as TS-1 is a highly selective material, isomorphously substituted Ti+ atoms within the framework endow with TS-1 (Figure 2).

The Ultem membrane modified with 2.5 wt. % GO showed good results in the sorption experiment. Due to the hydrophilic nature of GO, the resulting membranes appeared to be more hydrophilic with higher pure-water flux recovery ratio. GO incorporation caused a decrease in the contact angle and improved the antifouling ability [56]. The GO layer can provide faster electron/ion pathways through non-covalent π−π interactions between the GO surface and the benzyl ring of IBU (Figure 3) [65].

Finally, ALD silane modification of the PDMS surface membrane enhanced the sorption capacity of the membrane during the pertraction process. Bonding between IBU and adsorbed silane molecules stem from the coulombic interactions between oppositely charged parts of the molecules, resulting in weak bonding.

Figure 4 illustrates the combination of electrostatic and non-electrostatic interactions such as hydrogen bonding and hydrophobic interaction in the adsorption of IBU. The IBU carbocyclic groups interact through hydrogen bonding with the amine groups of ALD silane [64].

## 5. Conclusions

A wide set of membranes was extensively evaluated towards Ibuprofen separation from aqueous solution by adsorption and pertraction processes. In particular, twelve tailor-made membranes based on Matrimid, Ultem, and DAM/DABA and containing various amounts of zeolite or GO were compared to three commercial polymer membranes in terms of capacity for IBU sorption from aqueous solution. The highest capacity was reached with Matrimid + 20 wt. % TS-1 zeolite (0.58 mg/cm^2^), followed by Ultem + 2.5 wt. % GO (0.41 mg/cm^2^) and 6FDA-DAM:DABA 3:1 + 2.8 wt. % GO (0.38 mg/cm^2^). The optimal amount of filler for each type of tailor-made polymer membrane was determined. The commercial membranes were mechanically significantly more stable and therefore already suitable for pertraction experiments. This unique process of IBU elimination tripled the quantity of IBU removed from the feed during the time of testing. Atomic layer deposition of (3-aminopropyl)triethoxysilane on the best commercial membrane, PDMS, prevented from membrane saturation during the tested time and significantly improved the capacity of the material in the uptake of IBU from the feed compared to bare PDMS (53% instead of 23% of IBU in 16 days). A direct comparison with other research studies has not been considered due to the different nature of the analysis. The promising results of this method towards IBU elimination from water, with the high and predictive selectivity of a dense, hydrophobic polymer membrane such as PDMS open a new pathway that deserves attention from researchers. Further attention should be focused mainly on the mechanical stability of promising tailor-made membranes for pertraction of pollutants, on the enhancement of process duration, and improve permeation through the dense membranes.

## Data Availability

The data that support the findings of this study are available from the corresponding author upon reasonable request.

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
