# Peer review of "Removal of Ibuprofen from Water by Different Types Membranes"

_polymers, 2021, doi:10.3390/polym13234082_

Round 1

Reviewer 1 Report

Manuscript Number: polymers-1441883

Title: Removal of ibuprofen from water by different types of hydrophobic polymer membranes

Type: Research article

Recommendation: Major Revision

Comments to authors: The authors have presented an interesting study about Ibuprofen separation from water by adsorption and pertraction. The work can be considered for publication after addressing the following comments.

  • On page 5, line 150-151 author states that “before the experiment into round membranes of 3 or 3.8 cm diameter, then fixed between two parts of the disc with screws”. How did author ensured the mechanical stability of these membranes when screwed? Also did they check for any possible leak post tightening?

  • How can author explain the oscillatory behavior of Figure 3? Should the concentration be decreasing with time?

  • On page 7 line 221-222, authors states that “This may be due to better adhesion between Ti particles and the carbonyl groups of Matrimid.” How did author know about this better adhesion?

  • For Figure 9 authors are advised to present in terms of IBU rejection. That will give a much better idea and will be easier for readers to comprehend.

Reviewer 2 Report

Detailed comments:

  1. The English of the text should be checked
  2. The authors must be included new, relevant and more information about other membrane treatment processes (e.g., electrodialysis, reverse-electrodialysis, forward osmosis). Also, must be included the advantages and disadvantage of these processes. The following references should be included in the Introduction part to improve the quality of manuscript, because they provide relevant information:
  • Purification of H2SO4 of pickling bath contaminated by Fe(II) ions using electrodialysis process. Energy Procedia 2015, 74, 1418–1433.
  • Biopolymeric Membrane Enriched with Chitosan and Silver for Metallic Ions Removal, Polymers 2020, 12, 1792; doi:10.3390/polym12081792
  • Removal of Copper Ions from Simulated Wastewaters Using Different Bicomponent Polymer Membranes, Water, Air, & Soil Pollution, 225 (8), 2014, 2079
  • Performance of integrated membrane filtration and electrodialysis processes for copper recovery from wafer polishing wastewater. J. Water Process Eng. 2014, 4, p. 149–158.
  • Membrane Technologies in Wastewater Treatment: A Review. Membranes 2020, 10, 89.
  • Sci. 2020, 10, 7317; doi:10.3390/app10207317

  1. Please eliminate multiple references. After that, please check the manuscript thoroughly and eliminate ALL the lumps in the manuscript. This should be done by characterizing each reference individually and by mentioning 1 or 2 phrases per reference to show how it is different from the others and why it deserves mentioning. Multiple references are of no use for a reader and can substitute even a kind of plagiarism, as sometimes authors are using them without proper studies of all references used. In the case, each reference should be justified by it is used and at least short assessment provided.  
  2. At part 2.1. Sample preparation please indicate more information: value for temperature, for stirring overnight – must indicate the speed of stirred, magnetic stirred, indicate the time; also, for “1/1 acetonitrile/methanol” – indicate what means 1/1, molar rate?
  3. Comparison between the obtained results and measured in this study with other reported studies should be done and included for more clarity (indicate values not just number of reference).

Round 2

Reviewer 1 Report

Authors have addressed my concerns and the manuscript can be accepted now.

Author Response

Thank you.